# EDITROOM: LLM-PARAMETERIZED GRAPH DIFFUSION FOR COMPOSABLE 3D ROOM LAYOUT EDITING

**Kaizhi Zheng**[1]    **Xiaotong Chen**[2]    **Xuehai He**[1]    **Jing Gu**[1]    **Linjie Li**[3]
**Zhengyuan Yang**[3]    **Kevin Lin**[3]    **Jianfeng Wang**[3]
**Lijuan Wang**[3]    **Xin Eric Wang**[1]
[1]UC Santa Cruz    [2]University of Michigan, Ann Arbor    [3]Microsoft
{kzheng31,xwang366}@ucsc.edu

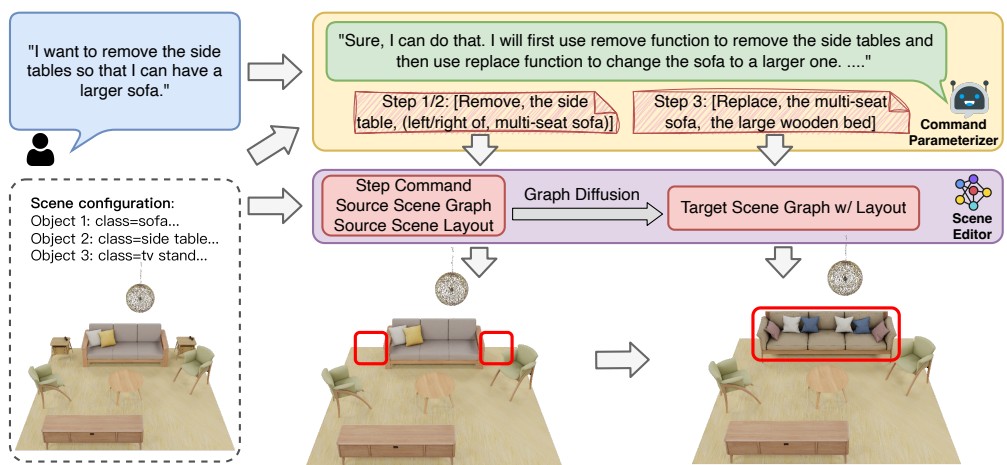

Figure 1: **Editing Pipeline with EditRoom.** EditRoom is a unified language-guided 3D scene layout editing framework that can automatically execute all layout editing types with natural language commands, which includes the command parameterizer for natural language comprehension and the scene editor for editing execution. Given a source scene and natural language commands, it can generate a coherent and appropriate target scene.

## ABSTRACT

Given the steep learning curve of professional 3D software and the time-consuming process of managing large 3D assets, language-guided 3D scene editing has significant potential in fields such as virtual reality, augmented reality, and gaming. However, recent approaches to language-guided 3D scene editing either require manual interventions or focus only on appearance modifications without supporting comprehensive scene layout changes. In response, we propose **EditRoom**, a unified framework capable of executing a variety of layout edits through natural language commands, without requiring manual intervention. Specifically, EditRoom leverages Large Language Models (LLMs) for command planning and generates target scenes using a diffusion-based method, enabling six types of edits: `rotate`, `translate`, `scale`, `replace`, `add`, and `remove`. To address the lack of data for language-guided 3D scene editing, we have developed an automatic pipeline to augment existing 3D scene synthesis datasets and introduced **EditRoom-DB**, a large-scale dataset with 83k editing pairs, for training and evaluation. Our experiments demonstrate that our approach consistently outperforms other baselines across all metrics, indicating higher accuracy and coherence in language-guided scene layout editing. Project website: https://eric-ai-lab.github.io/edit-room.github.io/

# 1 INTRODUCTION

Traditionally, editing 3D scenes requires manual intervention through specialized software like Blender (Community, 2024), which demands substantial expertise and considerable time for resource management. As a result, language-guided 3D scene editing has emerged as a promising technology for next-generation 3D software. To build an automated system capable of interpreting natural language and manipulating scenes, the system must be able to align complex, diverse, and often ambiguous language commands with various editing actions while also comprehending the global spatial structure of the scene. Additionally, the relatively small size of available 3D scene datasets presents a challenge for developing large-scale pretrained models necessary for fully automated, end-to-end language-guided scene editing.

Recently, there have been some works (Zhuang et al., 2023; Bartrum et al., 2024; Karim et al., 2023) focusing on leveraging pretrained image generation models to edit single object appearance inside the scene, but they fail to modify the layout of original scenes. Meanwhile, other approaches (Chen et al., 2023; Ye et al., 2023) further incorporate pretrained segmentation models to enable individual object manipulation. However, they require manual intervention to determine the edited object and editing type for any layout adjustments, like adding a new object or changing the object pose. Furthermore, these methods are all limited to executing a single editing step and are hard to deal with commands with multiple potential steps.

Therefore, we propose **EditRoom**, a unified framework that can execute all editing types by complex natural language commands without intermediate manual interventions. It consists of two main modules: the *command parameterizer* and the *scene editor*. The *command parameterizer* employs an pretrained LLM, specifically GPT-4o (OpenAI, 2024), to transform natural language commands into sequences of breakdown commands for six basic editing types on single object: `adding`, `removing`, `replacing`, `translating`, `rotating`, and `scaling`. These breakdown commands, along with the source scenes, are then fed sequentially into the *scene editor* for execution. To unify all editing types, we convert scenes into graph representations and construct *scene editor* as conditional graph generation models, where we take source scenes and text commands as conditions and train diffusion models to generate the target scene graphs with the layout. By conditioning on the user's natural language prompts, our model generates reasonable editing results that serve as a foundation for further refinement. Importantly, the scenes remain fully editable, allowing users to provide additional prompts for iterative improvements to achieve their desired outcome.

Another challenge is the lack of language-guided scene editing datasets, which constrains both training and evaluation of scene editing models. To enable the *scene editor* to accurately execute every basic editing type, we construct an automatic data generation pipeline and collect a synthetic scene editing dataset named **EditRoom-DB** for both training and evaluation, which includes approximately 83k editing pairs with commands. We implement several functions to simulate the single editing process on the existing 3D scene synthetic dataset, 3D FRONT (Fu et al., 2021a), which contains 16k indoor scene designs equipped with high-quality object models, and generate corresponding language commands using predefined templates. To mimic the human inputs, we employ LLMs to transform the templated commands into natural language forms, serving both as training material for our baselines and as test cases for single-operation evaluations.

In the experiments, we quantitatively assess the performance of EditRoom in scenarios with single-operation commands. From the results, we find that EditRoom outperforms other baselines in all metrics across different room types and editing types, which indicates higher precision and coherence in single-operation editing. Furthermore, we qualitatively evaluate EditRoom in scenarios involving multi-operation commands. We find that the model can successfully generalize to these scenarios even though we do not train the model on multi-operation data.

Our contributions are summarized as follows:

- We propose a new framework, named EditRoom, consisting of the *command parameterizer* and *scene editor*, which accepts scene inputs and can edit scenes using natural language commands by leveraging LLM for planning.
- We propose a unified graph diffusion-based module that serves as the *scene editor*, capable of executing every basic editing type, including `adding`, `removing`, `replacing`, `translating`, `rotating`, and `scaling`.

- We introduce the EditRoom-DB dataset with 83k editing pairs for the first language-guided 3D scene layout editing dataset by augmenting the existing 3D scene synthesis dataset.

- From the experiments, we demonstrate that EditRoom outperforms other baselines across all editing types and room types on single operation commands, and it can generalize to complex operation commands without further training.

## 2 RELATED WORK

**Language-Guided 3D Scene Generation** Language-guided 3D scene generation works claiming editing capabilities can be categorized into two main approaches. The first approach utilizes large language models to generate scene configurations and perform edits (Vilesov et al., 2023; Zhou et al., 2024b; Aguina-Kang et al., 2024). However, these methods are limited to editing scenes they generate and cannot accept external source scenes. The second approach trains diffusion-based models for text-conditional scene generation (Haque et al., 2023; Tang et al., 2023; Zhai et al., 2024). These methods often require manual masking for edits and have limited editing functionalities. For example, *InstructScene* (Haque et al., 2023) lacks support for operations like translation, rotation, and scaling, while *EchoScene* (Zhai et al., 2024), which directly take target scene graphs as input, only modifies object relations manually and does not process textual instructions. In contrast, EditRoom accepts existing scenes and supports free-form editing commands for 3D layouts. By leveraging diffusion models trained on clean data, EditRoom enables comprehensive edits, including translating, rotating, scaling, adding, and removing objects, without manual intervention.

**Language-guided 3D Scene Editing** Previously, some works (Ma et al., 2018) have explored the rule-based methods for language-guided 3D scene editing. Compared to these works, our diffusion-based method can generate diverse editing results, generalize across editing types. Recent language-guided 3D scene editing works mostly incorate neural field representations and pretrained image generation models for object appearance editing. Some works are mainly focusing on replacing single object (Zhuang et al., 2023; Bartrum et al., 2024; Karim et al., 2023) but failing to edit the layout. Other approaches (Chen et al., 2023; Ye et al., 2023) leverage pretrained segmantation models to obtain the individual object representation so that they can remove the objects by manually selecting the target objects. Besides, all these methods can only take one single-operation command at each interaction. Compared to these previous works, EditRoom can leverage LLM to deal with multi-operation natural language commands and automatically execute all editing types through a unified graph diffusion-based module.

**LLMs for 3D Scene Understanding** Recent works demonstrate that existing LLMs can facilitate 3D spatial reasoning. These works usually leverage the pretrained image caption models to convert 3D scenes into text descriptions and ask the LLM to generate navigation steps (Zhou et al., 2023; 2024a), provide room layout (Feng et al., 2024), or ground 3D objects (Yang et al., 2023; Hong et al., 2023; Huang et al., 2023). In our work, we are the first work to leverage LLM for natural language-guided 3D layout editing, where LLM takes source scenes in text format and breaks the natural language commands into basic editing operations.

**Diffusion Models for Graph** In recent years, denoising diffusion models have shown impressive generative capability in the graph generation (Liu et al., 2023a; Kong et al., 2023; Vignac et al., 2022). Compared to the previous VAE-based Verma et al. (2022) or GAN-based (Martinkus et al., 2022) models, diffusion-based models have advantages like stable training processes and generalizability to various graph structures. Some works have presented that graph diffusion-based models can be used in molecule generation (Guo et al., 2022; Akhmetshin, 2023), protein modeling (Zhang et al., 2022), 3D scene generation (Haque et al., 2023; Zhai et al., 2024), and etc. In this work, we first propose to use graph diffusion-based models for language-guided 3D scene layout editing.

## 3 THE EDITROOM METHOD

In this section, we introduce EditRoom, shown in Figure 1, comprising two primary modules: *Command Parameterizer* and the *Scene Editor*. Given a natural language command $C$ and source scene $S$, we aim to estimate the target scene $T$ with conditional distribution $q(T|S,C)$. Our *command parameterizer* takes the source scene $S$ and natural command $C$ to generate the breakdown commands

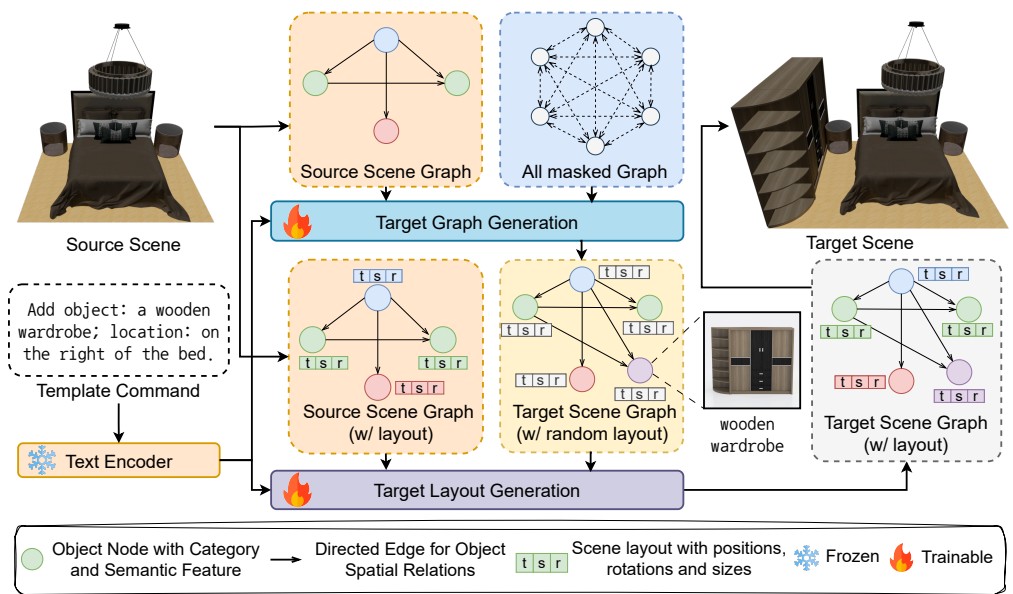

Figure 2: **Scene Editor Overview.** Scene Editor aims to provide accurate, coherent editing results according to the given source scene and language commands. It consists of two graph transformer-based conditional diffusion models. One diffusion model generates semantic target scene graphs. Another diffusion model can estimate accurate poses and size information for each object inside the generated target scene graphs. All diffusion processes are conditioned on the source scene and breakdown command.

$L$. Then, the *scene editor* conditions on breakdown commands $L$ to obtain the final target scene $T$, where the whole pipeline can be written as $q(T|S, C) = q(L|S, C) \times q(T|S, L)$. All objects inside source scenes and target scenes are retrieved from a high-quality 3D furniture dataset (Fu et al., 2021b).

## 3.1 LLM AS COMMAND PARAMETERIZER

In order to process open natural language commands, we use GPT-4o (OpenAI, 2024) to convert natural language command $C$ into a set of combinations of basic editing types with breakdown commands $L := \{l_j\}_{j=1}^{N_L}$, where $N_L$ is the number of breakdown commands. To cover the general manipulations on the scene, we design six basic editing operations:

- Rotate an object: [Rotate, Target Object Description, Angle]

- Translate an object: [Translate, Target Object Description, Direction, Distance]

- Scale an object: [Scale, Target Object Description, Scale Factor]

- Replace an object: [Replace, Source Object Description, Target Object Description]

- Add an object: [Add, Target Object Description, Target Object Location]

- Remove an object: [Remove, Target Object Description]

When the target object is not unique, we ask the LLM to use another unique object as a reference to describe the spatial relation. During the inference phase, we prompt the LLM with attributes of objects within the source scene along with the natural language command, tasking the model to analyze the scene and delineate basic editing operations through breakdown commands in specified formats. The attributes include categories, locations, sizes, rotations, and object captions. Detailed descriptions of the full prompt and examples are provided in Figure 6 of the appendix.

## 3.2 GRAPH DIFFUSION AS 3D SCENE EDITOR

Given the breakdown command $l$ and source scene $S$, our objective is to determine the conditional target scene distribution $q(T|S, l)$. Drawing inspiration from recent advancements in language-guided 3D scene synthesis (Lin & Yadong, 2023), we transform scenes into semantic graphs and employ a graph transformer-based conditional diffusion model to learn the conditional target scene graph distribution, as depicted in Figure 2. Our approach involves two key graph transformer-based diffusion models: the *Target Graph Diffusion*, which generates object shapes and their spatial relations as graphs, and the *Target Layout Diffusion*, which computes the final layout of the target scene. To reduce the alignment challenges between the 3D scene distribution and language, all commands are encoded using the text encoder of CLIP-ViT-B-32 (Radford et al., 2021).

**Scene Graph Representation** Each scene is represented as a combination of a layout $B$ and a scene graph $\mathcal{G}$ (Lin & Yadong, 2023). The layout $B$ encapsulates the position, size, and orientation of each object, while the scene graph $\mathcal{G}$ encodes additional high-level semantic information. Formally, a semantic scene graph $\mathcal{G} := (V, E)$ comprises nodes $v_i \in V$, where each $v_i$ corresponds to an object $o_i$ with high-level attributes. Directed edges $e_{ij} \in E$ represent spatial relationships: ["in front of", "behind", "right of", "left of", "closely in front of", "closely behind", "closely right of", "closely left of", "above", "below"], connecting the $i$-th object to the $j$-th object, where "closely" means the distance between two object centers are less than 1 meter. Each node $v_i$ is characterized by a discrete category $c_i$ and continuous semantic features $f_i$, derived from a pretrained multimodal-aligned point cloud encoder, OpenShape (Liu et al., 2024c), which features a 1280-dimensional representation space.

**Target Graph Diffusion** In this stage, we aim to learn target scene graphs $\mathcal{G}_{tg}$ by giving source scene graphs $\mathcal{G}_s$ and language commands $l$ through a discrete graph diffusion model $\varepsilon_g$, where $\mathcal{G}_{tg}$ includes category $C_{tg}$ and semantic features $F_{tg}$ for each node and the edges $E_{tg}$ for object relative relations. Since high-dimensional object semantic features ($d = 1280$) are too complicated to learn from limited data, we use a VQ-VAE model (Lin & Yadong, 2023; Wang et al., 2019) to compress them into low-dimensional features $z \in \mathbb{R}^{n_f \times d_Z}$, which consists of $n_f$ vectors extracted from a learned codebook $Z \in \mathbb{R}^{K_f \times d_Z}$ by a sequence of feature indices $f_{idx} := \{1, ..., K_f\}^{n_f}$, where $K_f$ and $d_Z$ are the size and dimension of codebook. Then, we use the feature indices to replace the original object semantic features as targets for training, denoted as $\hat{F}$. Therefore, $\mathcal{G}_{tg} = (C_{tg}, \hat{F}_{tg}, E_{tg})$ and $\mathcal{G}_s = (C_s, \hat{F}_s, E_s)$, and our goal is to learn the conditional distribution $q(\mathcal{G}_{tg}|\mathcal{G}_s, l)$. During the training process, at timestep $t$, the gaussian noises are added to the $\mathcal{G}_{tg}$ to get $\mathcal{G}_{tg}^t$, and the model $\varepsilon_g$ aims to reconstruct $\mathcal{G}_{tg}^0$ by conditioning on $\mathcal{G}_s$ and $l$. To add the conditions, we concatenate each element of source scene graphs into noisy target scene graphs as context and use cross-attention layers to incorporate language features. The loss function can be written as:

$$L_g := \mathbb{E}_{q(\mathcal{G}_{tg}^0)}[\sum_{t=2}^{T} L_{t-1} - \mathbb{E}_{q(\mathcal{G}_{tg}^1|\mathcal{G}_{tg}^0)}[\log p_{\varepsilon_g}(\mathcal{G}_{tg}^0|\mathcal{G}_{tg}^1, \mathcal{G}_s, l)]] \tag{1}$$

$$L_{t-1} := D_{KL}[q(\mathcal{G}_{tg}^{t-1}|\mathcal{G}_{tg}^t, \mathcal{G}_{tg}^0)||p_{\varepsilon_g}(\mathcal{G}_{tg}^{t-1}|\mathcal{G}_{tg}^t, \mathcal{G}_s, l)] \tag{2}$$

where $D_{KL}$ indicates the KL divergence.

**Target Layout Diffusion** In this stage, we aim to estimate the target scene layout $B_{tg}$ using another graph diffusion model $\varepsilon_b$, conditioning on target scene graph $\mathcal{G}_{tg}$, source scene graph $\mathcal{G}_s$, source layout $B_s$, and language command $l$. The target scene layout $B_{tg} \in \mathbb{R}^{M \times 8}$ consists of position $T_{tg} \in \mathbb{R}^{M \times 3}$, size $S_{tg} \in \mathbb{R}^{M \times 3}$, and rotation $R_{tg} \in \mathbb{R}^{M \times 2}$. During the training process, gaussian noises $\epsilon$ will be added to the target layout, and the layouts are encoded into the node features by MLP layers. Similar to the *Target Graph Diffusion*, we concatenate the source scene graph and source layout to the target scene graph as context and corrupted target layout. The language features are incorporated through cross-attention layers. The objective target is to estimate the added noises at each time step. The loss function can be written as:

$$L_b := \mathbb{E}_{B_{tg}^0, t, \epsilon}[||\epsilon - \varepsilon_b(B_{tg}^t, t, \mathcal{G}_{tg}, \mathcal{G}_s, B_s, l)] \tag{3}$$

**Inference Process** During the inference phase, the first step consists of transforming the source scene into a scene graph $\mathcal{G}_s$ and a corresponding layout $B_s$. Subsequently, the *Target Graph Generation* model predicts the target scene graph $\mathcal{G}_{tg}$, conditioned on the source scene graph $\mathcal{G}_s$ and the

Table 1: **EditRoom-DB dataset statistics.** We collect around 83k training data across all room types and 500 test data for each room type.

| Types | Train | | | Test | | |
|---|---|---|---|---|---|---|
| | Bedroom | Dining room | Living room | Bedroom | Dining room | Living room |
| Translate | 8.6k | 3.2k | 2.7k | 808 | 253 | 250 |
| Rotate | 4.0k | 1.3k | 1.3k | 804 | 250 | 252 |
| Scale | 12.7k | 4.5k | 3.9k | 805 | 251 | 252 |
| Add | 8.9k | 3.4k | 2.8k | 747 | 232 | 240 |
| Remove | 8.8k | 3.3k | 2.8k | 816 | 260 | 253 |
| Replace | 6.8k | 2.2k | 2.1k | 820 | 254 | 253 |
| Total | 49.8k | 17.9k | 15.6k | 4800 | 1500 | 1500 |

language command $l$. This is followed by the *Target Layout Generation* model, which computes the target layout $B_{tg}$, leveraging all available variables as inputs. The final step in constructing the target scene, denoted as $T := (\mathcal{G}_{tg}, B_{tg})$, involves retrieving the object meshes based on the estimated object features and arranging them according to the generated layout. This systematic approach enables the dynamic generation of scenes that are aligned with verbal instructions, ensuring that the resulting scenes accurately represent the specified conditions.

## 4 THE EDITROOM-DB DATASET

To facilitate comprehensive training and evaluation of our framework, we present the **EditRoom-DB** dataset, designed to support a wide range of basic 3D scene editing operations. The dataset is generated through an automated data augmentation pipeline that produces editing pairs based on object-level modifications applied to scenes from the 3D-FRONT dataset (Fu et al., 2021a). We utilize scenes from the bedroom, dining room, and living room categories and enhance them with high-quality object models from the 3D-FUTURE dataset (Fu et al., 2021c) to simulate real-world editing workflows.

Our data generation pipeline supports a variety of editing operations, including `Add and Remove Objects`, `Pose and Size Changes`, and `Object Replacement`. Each modification produces a new scene paired with a detailed textual description of the changes made, following a predefined template. For each scene, objects are selected randomly and modified iteratively through basic editing operations. In the case of `Add and Remove Objects`, the modified scene serves as the target for the operation, with the original scene acting as the source. For `Pose and Size Changes`, random adjustments are applied to selected objects, and collision checking ensures that the final scenes are free from object overlap. Similarly, in `Object Replacement`, new objects from the same category are substituted for existing ones, with collision checks ensuring high-quality outputs. More details about automatic pipelines can be found in Appendix C.

To create diverse and realistic language instructions for the editing tasks, we first employ the multimodal understanding model LLAVA-1.6 (Liu et al., 2024b;a; 2023b), which captions front-view images of the objects in the scenes. Then, these object captions are used to construct the template-based commands. The template-based commands are then transformed into natural language commands using GPT-4o, making the dataset suitable for training and testing language-guided scene editing models. Detailed prompts and additional examples are shown in Figure 7 in the appendix.

The resulting dataset consists of approximately 83,000 training samples and 7,800 test samples (randomly sampled across editing types for each room type). Table 1 provides a detailed breakdown of the dataset statistics across the three scene categories. Each sample includes a source scene, an edited target scene, and corresponding language commands. This comprehensive dataset enables the development and evaluation of robust models capable of performing various scene editing tasks. By simulating realistic workflows and providing detailed text commands, EditRoom-DB serves as a valuable resource for advancing language-guided 3D scene editing.

Table 2: **Performance on single operation with different room types.** From the table, we can find EditRoom outperforms baselines among all room types, which indicates that our methods can provide more accurate and coherent editing across room types.

| Model | Bedroom | | | | Dining room | | | | Living room | | | |
|---|---|---|---|---|---|---|---|---|---|---|---|---|
| | IOU (↑) | S-IOU (↑) | LPIPS (↓) | CLIP (↑) | IOU (↑) | S-IOU (↑) | LPIPS (↓) | CLIP (↑) | IOU (↑) | S-IOU (↑) | LPIPS (↓) | CLIP (↑) |
| DiffuScene-N | 0.6220 | 0.6125 | 0.1431 | 0.9415 | 0.4473 | 0.4321 | 0.1921 | 0.9254 | 0.4332 | 0.4161 | 0.1810 | 0.9236 |
| SceneEditor-N | 0.7055 | 0.6942 | 0.1261 | 0.9510 | 0.5262 | 0.5091 | 0.1557 | 0.9366 | 0.4618 | 0.4498 | 0.1738 | 0.9329 |
| EditRoom | **0.7342** | **0.7236** | **0.1090** | **0.9597** | **0.5360** | **0.5196** | **0.1460** | **0.9460** | **0.4710** | **0.4629** | **0.1616** | **0.9446** |

## 5 EXPERIMENTS

### 5.1 EXPERIMENTAL SETUP

**Baselines** To the best of the authors' knowledge, EditRoom is the first work that can automatically execute all editing types with natural language commands. Therefore, we construct two baselines by modifying language-guided 3D scene synthesis methods for comparisons: DiffuScene-N and SceneEditor-N:

- DiffuScene-N: DiffuScene-N is modified from the language-guided 3D scene synthesis work, DiffuScene (Tang et al., 2023), which includes a UNet-based diffusion model to generate scene layout. To enable it with language-guided scene editing ability, we leverage their scene completion pipeline by incorporating the source scene as context for the diffusion process. During the training and testing, the model directly conditions natural commands for target scene layout generation.
- SceneEditor-N: To test our generalization ability, we experiment with another setting, where the *scene editor* directly trains on the natural commands got from the GPT-4o. During the inference time, the model conditions the natural commands and generates the final scenes.

We would like to emphasize that not all language-guided 3D scene synthesis methods can be modified as 3D scene layout editing methods. There are at least two prerequisites: (1) there is a way to easily convert the source scene into the required intermediate representations; (2) the scene representations have the capability to execute all editing manipulations. Some LLM-relevant 3D scene generation models (Vilesov et al., 2023; Zhou et al., 2024b; Aguina-Kang et al., 2024; Hu et al., 2024) can generate 3D scenes based on descriptions, but source scenes are hard to convert into their intermediate representations, like program language (Aguina-Kang et al., 2024) and blender codes (Hu et al., 2024).

**Metrics** To evaluate the models' performance, we utilize four metrics: IOU, S-IOU, LPIPS (Zhang et al., 2018), and CLIP (Radford et al., 2021) scores. The IOU scores are calculated by determining the 3D Intersection Over Union (IOU) between each object in the generated and target scenes, selecting pairs with the highest 3D IOU values. The S-IOU represents the semantic-weighted 3D IOU, where semantic similarities between matching objects are calculated using Sentence BERT (S-BERT) (Reimers & Gurevych, 2019) based on their captions. For visual evaluation, we render both the generated and target scenes from 24 fixed camera views. Visual similarity is assessed using the LPIPS metric for pixel similarity, and semantic similarity is evaluated using the CLIP image encoder (CLIP-ViT-B32).

### 5.2 RESULTS

**Single-operation Commands** To assess model performance on single operations, we test our model and baselines using the EditRoom-DB test set, which contains 500 samples per room type, with language commands generated by GPT-4o. Quantitative results are depicted in Tables 2 and 3, and qualitative outcomes are illustrated in Figure 3. Table 2 indicates that EditRoom consistently outperforms other baselines across all room types, with notably superior performance in bedrooms. According to Table 3, EditRoom also excels across all editing types. Comparisons between EditRoom and SceneEditor-N reveal that template-based instructions can simplify the learning process by more effectively aligning language commands with scene changes. Moreover, the LLM (GPT-4o) demonstrates a successful bridge between natural language and template commands. SceneEditor-

Table 3: **Performance on single operations with different editing types.** From the table, we can notice EditRoom provides better editing results across all basic editing types.

| | Translate | | | | Rotate | | | |
|---|---|---|---|---|---|---|---|---|
| Model | IOU (↑) | S-IOU (↑) | LPIPS (↓) | CLIP (↑) | IOU (↑) | S-IOU (↑) | LPIPS (↓) | CLIP (↑) |
| DiffuScene-N | 0.5363 | 0.5279 | 0.1715 | 0.9484 | 0.6010 | 0.5913 | 0.1307 | 0.9511 |
| SceneEditor-N | 0.5983 | 0.5905 | 0.1534 | 0.9504 | 0.6655 | 0.6556 | 0.1202 | 0.9560 |
| EditRoom | **0.6148** | **0.6078** | **0.1388** | **0.9584** | **0.6764** | **0.6673** | **0.1067** | **0.9610** |

| | Scale | | | | Replace | | | |
|---|---|---|---|---|---|---|---|---|
| Model | IOU (↑) | S-IOU (↑) | LPIPS (↓) | CLIP (↑) | IOU (↑) | S-IOU (↑) | LPIPS (↓) | CLIP (↑) |
| DiffuScene-N | 0.6057 | 0.5947 | 0.1250 | 0.9566 | 0.5994 | 0.5754 | 0.1488 | 0.9305 |
| SceneEditor-N | 0.6754 | 0.6650 | 0.1105 | 0.9603 | 0.6451 | 0.6193 | 0.1418 | 0.9389 |
| EditRoom | **0.6881** | **0.6788** | **0.1008** | **0.9647** | **0.6549** | **0.6305** | **0.1384** | **0.9419** |

| | Add | | | | Remove | | | |
|---|---|---|---|---|---|---|---|---|
| Model | IOU (↑) | S-IOU (↑) | LPIPS (↓) | CLIP (↑) | IOU (↑) | S-IOU (↑) | LPIPS (↓) | CLIP (↑) |
| DiffuScene-N | 0.5312 | 0.5198 | 0.1799 | 0.9263 | 0.4626 | 0.4511 | 0.1877 | 0.9281 |
| SceneEditor-N | 0.5863 | 0.5745 | 0.1620 | 0.9355 | 0.6160 | 0.6065 | 0.1290 | 0.9489 |
| EditRoom | **0.6001** | **0.5887** | **0.1588** | **0.9410** | **0.6399** | **0.6316** | **0.1150** | **0.9583** |

Table 4: **Ablation on different condition types on the bedroom.** From the table, we can show that incorporating source information as context with the self-attention (our design) instead of the cross-attention mechanism can significantly improve model performance.

| Model | IOU (↑) | S-IOU (↑) | LPIPS (↓) | CLIP (↑) |
|---|---|---|---|---|
| EditRoom (Concat-Text) | 0.5855 | 0.5746 | 0.1432 | 0.9488 |
| EditRoom (Original) | **0.7342** | **0.7236** | **0.1090** | **0.9597** |

N outperforms DiffScene-E across all metrics and editing types, suggesting that our graph-based diffusion method yields more coherent and accurate editing results compared to the UNet-based approach. Thus, EditRoom provides more precise and coherent atomic editing operations from natural language commands than its counterparts.

Analysis across different room types shows that all models perform better as the average number of objects in rooms decreases, highlighting potential for improvements in larger, more complex scenes. Evaluating different editing operations reveals that translating, adding, and removing operations score lower on IOU, demanding stronger spatial reasoning. Meanwhile, replacing and adding operations yield lower CLIP scores, indicating a need for better alignment between object descriptions and their semantic features. This underscores the potential for further enhancement of models' spatial reasoning and object alignment capabilities.

**Multi-operation Commands** To demonstrate the generalization capabilities of EditRoom, we manually designed several test prompts that combine multiple atomic operations, and we assessed each model's performance qualitatively. Figure 4, shows that EditRoom provides more coherent and appropriate responses than the baseline models. For instance, the command in the first row requests a bed replacement and the addition of a wardrobe. EditRoom successfully interprets the natural language command and translates it into the corresponding atomic operations using an LLM, whereas other baseline models misinterpret the command and perform incorrect operations such as translation. These outcomes highlight the challenges of directly training models on natural language commands for compositional editing tasks. EditRoom, by contrast, effectively executes complex editing operations through strategic LLM planning.

**Ablation on Condition Types** To validate our model design, we experimented with an alternative conditioning approach, where a graph transformer encodes the source scene into a sequence of vectors that are then concatenated with text features. These combined features are incorporated into the cross-attention layers of our graph diffusion process. We specifically tested this method on the bedroom scene type, with results shown in Table 4. The table indicates a significant decrease in model performance, both in terms of layout accuracy and visual coherence. This outcome suggests that

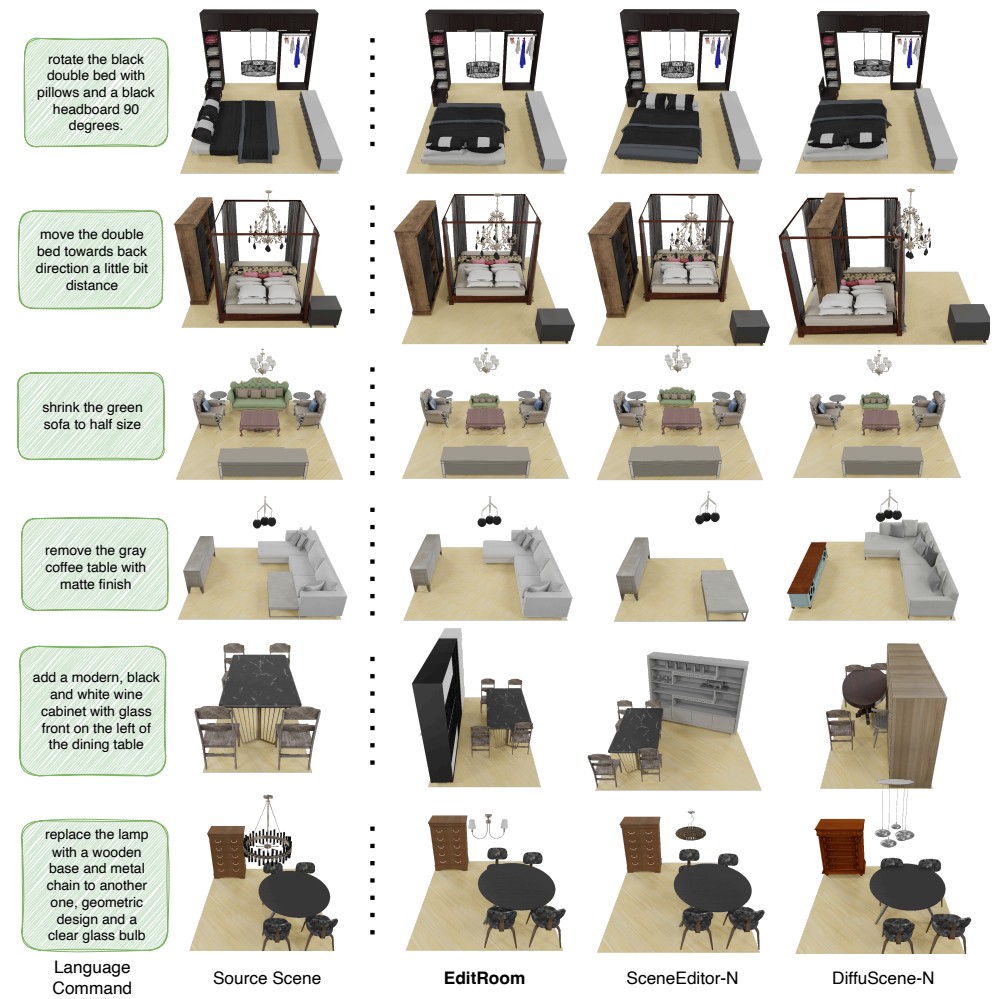

Figure 3: **Qualitative results on single-operation commands.** The left column is the source scene with single operation commands for each basic editing type. From the examples, we can find that EditRoom can provide more coherent and appropriate editing operations across all editing types.

Table 5: **Ablation on different text encoders on the bedroom.** Due to the limited size of training data, we find using the larger text encoder with high-dimensional features induces decreasing performance on editing, which indicates further exploration with 3D editing data generation.

| Model | IOU (↑) | S-IOU (↑) | LPIPS (↓) | CLIP (↑) |
|---|---|---|---|---|
| EditRoom (OpenCLIP-ViT-bigG-14) | 0.6882 | 0.6743 | 0.1305 | 0.9488 |
| EditRoom (Original) | **0.7342** | **0.7236** | **0.1090** | **0.9597** |

utilizing source scene information as the context for self-attention layers, rather than as conditions for cross-attention, yields better results.

**Ablation on Text Encoders** In an exploration of text encoder options, we replaced the CLIP-ViT-B32 text encoder (512 feature dimensions) with a larger pretrained encoder, OpenCLIP-ViT-bigG-14 (1280 feature dimensions), used by OpenShape to align with object semantic features—consistent with the object features in our models. We conducted tests on the bedroom test set, with outcomes detailed in Table 5. The results indicate that the model equipped with the larger text encoder underperforms compared to the one using the original encoder. We attribute this decrease in performance to the limited size of our training dataset. Given that our diffusion models are trained from

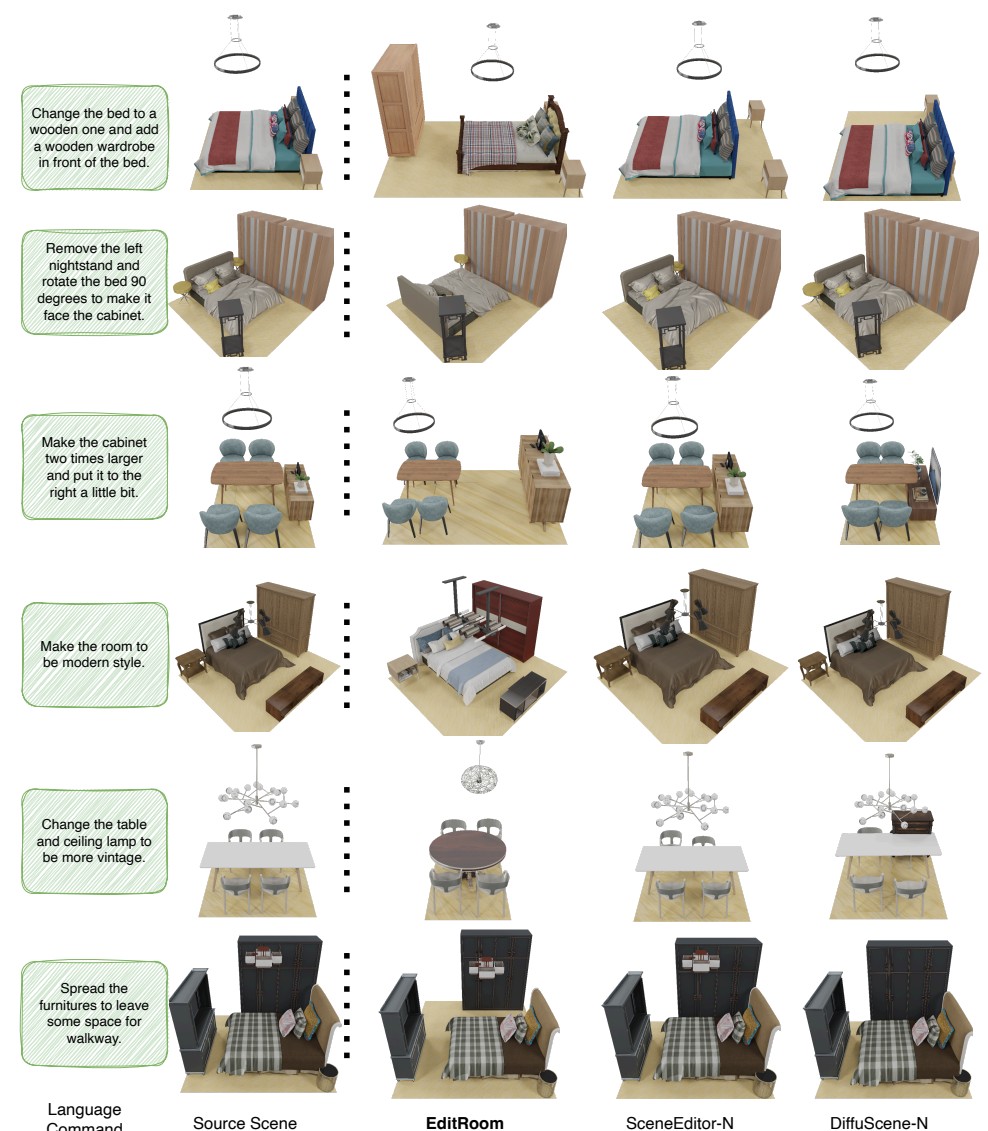

Figure 4: **Qualitative results on multi-operation commands.** The left column is the source scene with multi-operation operation commands. From the figure, we can find the EditRoom can successfully generalize to complex natural language commands with multiple operations without further training on the multi-operation operation data, while baselines fail to execute coherent editing.

scratch, they require more data to effectively align with higher-dimensional features ($d = 512$ vs $d = 1280$). This finding underscores the need for further exploration into constructing larger scene editing datasets.

## 6    CONCLUSION

In this work, we introduce EditRoom, a language-guided 3D room layout editing method. EditRoom incorporates a graph diffusion-based scene editor that facilitates unified basic editing operations, and it utilizes an LLM for natural language planning. Our experiments demonstrate that EditRoom can effectively execute appropriate edits for both single and complex operations. We believe this work will inspire further research into language-guided 3D scene layout editing.

**Limitation**   Since EditRoom leverages the LLM for the command parameterizer, its performance is contingent upon the LLM's capability in 3D scene understanding and natural command comprehension. This dependency may lead to the generation of erroneous commands that prompt the scene editor to execute potentially problematic operations, such as collisions. However, because the training data predominantly consist of collision-free samples, there is an inherent trade-off between adhering strictly to the commands and avoiding collisions. If the commands deviate significantly from typical scenarios—such as moving an object 100 meters away—the model might instead perform a similar action that falls within the observed training distributions.

**Ethics Statement**   Our work presents a framework for automated 3D scene editing guided by natural language. The primary focus of this research is to advance technical capabilities in scene manipulation. We do not anticipate any ethical concerns or negative societal impacts arising from this work. All datasets used in our research are synthetic and publicly available, and no personally identifiable information or human-related data were involved.

**Reproducibility Statement**   All language commands are encoded through the pretrained CLIP-ViT-B32 text encoder. Each graph diffusion model includes a five-layer graph transformer model with 512 hidden dimensions and 8 attention heads. Training is conducted using the AdamW optimizer over 300 epochs, with a batch size of 512 and a learning rate of $2 \times 10^{-4}$. All models are individually trained and tested on each room type. For EditRoom, template commands are employed for the scene editor during training, whereas other baseline models utilize natural language commands generated by GPT-4o. During testing, all models receive natural language commands as input. We set the node number of denoise graphs to the maximum object number for different room types (bedroom = 12, living room = dining room = 21). During training, we pad the objects with zeros to facilitate the batch process.

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

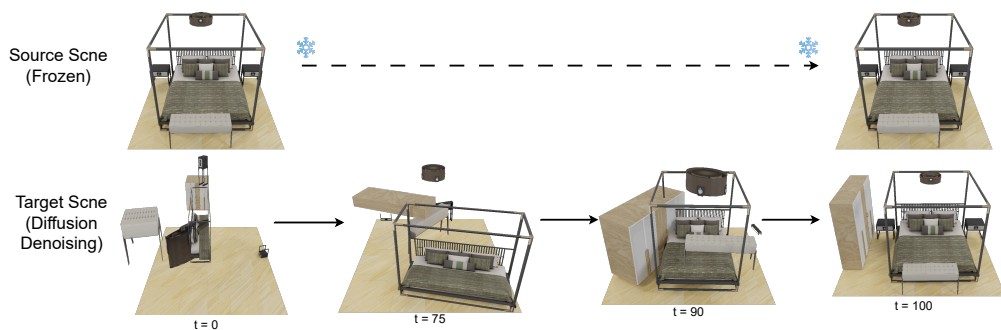

Instruction: "Add a wardrobe to the left of the bed."

Figure 5: **Visualization of layout diffusion denoising process.** The whole diffusion process is conditioned on both source scene and language commands. At the beginning of the process, the target scene layout starts from random noises. After the iterative diffusion denoising process, the target scene layout becomes coherent to source scene and command.

## A  LLM AS COMMAND PLANNER

(Referred by Section 3.1) A detailed dialog between user and LLM (GPT-4o) is shown in Figure 6.

## B  DIFFUSION DENOISING PROCESS

In order for a better illustration of target scene layout generation process, we visualize intermediate steps during the diffusion denoising process for an adding operation example, shown in Fig. 5. At the start of the layout diffusion process, the poses and scales of all objects inside the target scene begin as random noise. As the diffusion process progresses, each object's properties are iteratively refined and placed into an appropriate configuration that aligns with the source scene and the given commands. This iterative refinement highlights the strength of the diffusion model in handling scene edits cohesively. By adopting a unified approach, the diffusion model is capable of jointly generating the entire scene layout for all editing types, ensuring consistency and coherence across diverse operations.

## C  EDITROOM-DB PIPELINE DETAILS

(Referred by Section 4) A detailed example of using LLM to generate natural language description from template command is shown in Figure 7.

**Add and Remove Objects**   Removing each object in the scene separately could generate the modified scenes as the result after removal compared to the original scene. Conversely, the original scene could be treated as the result after object addition. The formatted editing description will be 'Add/Remove [object description]'. In order to consider the location of the addition and potential multiple objects in the scene, we will add the relative location description with the closest unique object in the scene, like 'location: [relative description] [reference object description]'.

**Pose and Size Changes**   We can similarly repeat the pose change operation for every object in the scene as add/remove. Specifically, we design three operations: translation, rotation, and scaling. For translation, we create random translations as the mix of distances, sampled from 0.1 meters to 1.5 meters with step 0.1, and directions, sampled along the two axes directions (front/back and left/right). Then, collision checking is done for every translated object until we find a collision-free sample. The translation will be skipped if all the samples fail in collision checking. The formatted editing description will be 'Move object towards the front/back/left/right direction for [distance] : [object description]'

Similarly, we create random rotation angles as the mix of uniform direction samples, clockwise or counterclockwise, and random values between $15 - 180$ degrees with the step of 15 degrees, and check collision for each sample. The check stops when we find a collision-free sample or all samples fail the checking. The formatted editing description will be 'Rotate object [angle] degrees : [object description]'

For scaling, we separate it as shrinking and enlarging. The scaling factor is randomly generated between 0.5-0.8 or 1.2-1.5. The scaling factor uniformly applies to three dimensions. Since shrinking won't cause a collision with other objects, it can always result in a successfully modified scene. For enlarging, if collision checking fails on all trials, the enlarging is skipped. Otherwise, we save the largest collision-free scaling factor. The formatted editing description will be 'Shrink/Enlarge object by [scale_factor] times : [object description]'

**Object Replacement**   For the replace operation, we access an object dataset with semantic class labels and 3D meshes. The system will retrieve several objects from the dataset with the same class label as the replaced object, and check their collision with other existing objects in the scene. If none of the objects could be placed without collision, we randomly select one object and shrink its bounding box to be equal or smaller than the replaced object to avoid collision. The formatted description is 'Replace source with target : [source object description] to [target object description]'.

**Collision Detection Module**   The objects are abstracted as 3D bounding boxes and further decomposed into 2D bounding boxes on a horizontal plane and vertical range, as the objects can only rotate about the vertical axis. Then, the two objects are only in collision if their 2D bounding boxes overlap and their vertical ranges overlap. For 2D bounding box collision detection, we apply the Separating Axis Theorem Huynh (2009) to determine if the boxes intersect.

**System Prompt**

Imagine you are an indoor room designer and you are using provided API to control the 3D models in the scene.
Given one scene configuration and a command to edit the scene, you should use the provided APIs to do planning and achieve the target.
All sizes and centroids in scene configurations are in meters. The angles are defined in degrees. The dimension sequence is [x,y,z]. Vertical angles are the angles along the y-axis.
Sizes are the half lengths of the bounding box along the x, y, and z axes when the vertical angle is zero.
We define +x/-x as the right/left direction, +y/-y as the up/down direction, and +z/-z as the front/back direction.
Positive angles are counterclockwise, and negative angles are clockwise.

APIs:
1. Rotate an object: [Rotate, Target Object Description, Angle :(degrees)]
2. Translate an object: [Translate, Target Object Description, Direction :(x/y/z), Distance :(meters)]
3. Scale an object: [Scale, Target Object Description, Scale Factor]
4. Replace an object: [Replace, Source Object Description, Target Object Description]
5. Add an object: [Add, Target Object Description]
6. Remove an object: [Remove, Target Object Description]

Matters needing attention:
1. If there are multiple same objects in the scene and the command is related to the object, you should refer to the object locations.
When you refer to the object locations, you should this format: (Relative Description, Relative Object Description).
When you use add or remove command, you should refer to the object locations.
Relative Description: [left, right, in front of, behind, above, below, closely left, closely right, closely in front of, closely behind]. 'closely' means the distance between two object centroids are less than 1 meters in x-z plane.
For example, if you want to add a chair in front of the table, you should use the format: [Add, Chair, (in front of, Table)].
2. Translate, rotate, and scale commands should be executed in the order of scale, rotate, and translate.
Consider the protential collision between objects when you execute the commands.
3. When you scale an object, the object should be scaled uniformly. Scale factor should one float number.
4. Replace object will only replace the object with the same class. Replace command will only change the object appearance, not the object poses and sizes.
5. If Translate/Rotate/Scale commands can achieve the target, you should not use Replace/Add/Remove commands.
6. If image is provided, you should use the image to help you understand the scene.
7. Attempt to use the minimum number of commands to achieve the target.
8. If you want to remove and add the object within the same class, you should use the replace command.
9. Object descriptions should be detailed descriptions instead of class names. You can imagine the object descriptions if the object is not in the scene.
10. All apis should be able to converted to a list of strings and numbers, which can be directly processed by json.loads()

For example:
1. If you want to rotate a chair 90 degrees and there is only one chair in the scene, you should use the format: ['Rotate', 'chair', 90].
2. If you want to add a chair in front of the table, you should use the format: ['Add', 'chair', ('in front of', 'table')].
3. If you want to remove a chair, you should use the format: ['Remove', 'chair'].
4. If you want to replace a metal chair with a wooden one and this chair on the left of the bed, you should use the format: ['Replace', 'mental chair', 'wooden chair', ('left', 'bed')].

Think about it step by step. Summarize the used apis at the end by lines. The final output format should be ***api1;api2;...***.

**User Input**

[Scene configurations]:
Object 0: {"class": "double bed", "size": [1.01, 0.39, 1.08], "vertical angle": -90, "centroid": [3.22, 0.0, -2.48],
"description": "the double bed is a modern, minimalist design with a white color scheme and a simple, clean appearance."}
Object 1: {"class": "nightstand", "size": [0.28, 0.24, 0.22], "vertical angle": 0, "centroid": [2.05, 0.0, -4.52],
"description": "the nightstand is a modern, dark wood piece with a sleek, minimalist design."}
Object 2: {"class": "nightstand", "size": [0.33, 0.33, 0.23], "vertical angle": 0, "centroid": [4.54, 0.0, -3.35],
"description": "the nightstand is a simple, white, two-drawer piece with a smooth finish and a small, round knob on each drawer."}
Object 3: {"class": "wardrobe", "size": [1.04, 1.02, 0.32], "vertical angle": 90, "centroid": [1.51, 0.0, -2.48],
"description": "the wardrobe is a large, white, traditional style piece of furniture with a decorative top and multiple doors."}
Object 4: {"class": "desk", "size": [1.0, 0.38, 0.3], "vertical angle": -90, "centroid": [5.6, 0.0, -2.47],
"description": "the desk is a modern, minimalist design with a sleek, rectangular shape and a matte gray finish."}
[Command]:"replace the nightstand near the white cabinet with the same model of another nightstand near the black table,
then move it closer to the bed corner to be symmetric with another one."

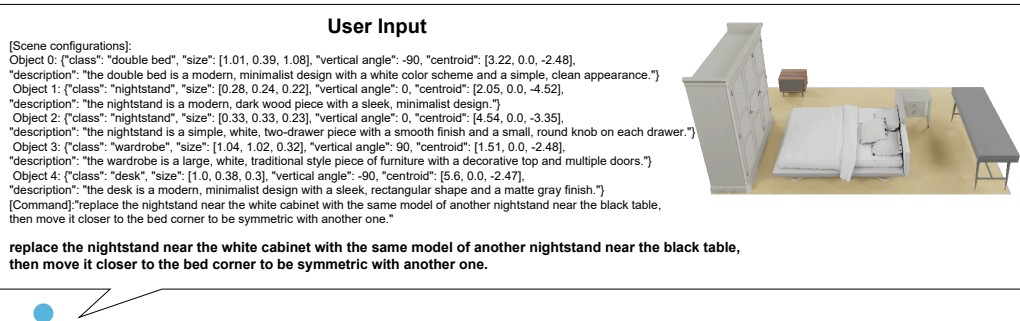

**replace the nightstand near the white cabinet with the same model of another nightstand near the black table,
then move it closer to the bed corner to be symmetric with another one.**

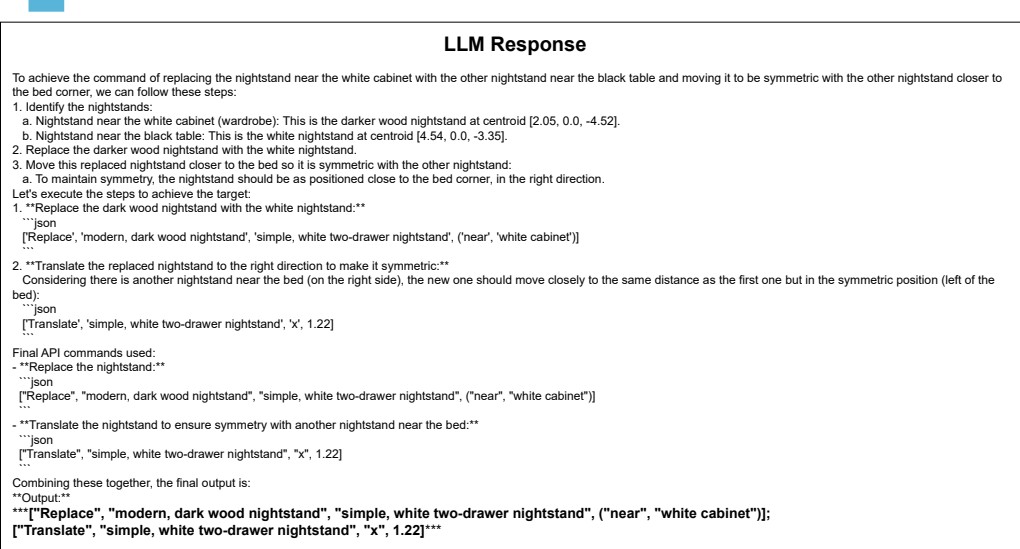

**LLM Response**

To achieve the command of replacing the nightstand near the white cabinet with the other nightstand near the black table and moving it to be symmetric with the other nightstand closer to the bed corner, we can follow these steps:
1. Identify the nightstands:
   a. Nightstand near the white cabinet (wardrobe): This is the darker wood nightstand at centroid [2.05, 0.0, -4.52].
   b. Nightstand near the black table: This is the white nightstand at centroid [4.54, 0.0, -3.35].
2. Replace the darker wood nightstand with the white nightstand.
3. Move this replaced nightstand closer to the bed so it is symmetric with the other nightstand closer to the bed corner:
   a. To maintain symmetry, the nightstand should be as positioned close to the bed corner, in the right direction.
Let's execute the steps to achieve the target:
1. **Replace the dark wood nightstand with the white nightstand:**
   ```json
   ['Replace', 'modern, dark wood nightstand', 'simple, white two-drawer nightstand', ('near', 'white cabinet')]
   ```
2. **Translate the replaced nightstand to the right direction to make it symmetric:**
   Considering there is another nightstand near the bed (on the right side), the new one should move closely to the same distance as the first one but in the symmetric position (left of the bed):
   ```json
   ['Translate', 'simple, white two-drawer nightstand', 'x', 1.22]
   ```
Final API commands used:
- **Replace the nightstand:**
   ```json
   ["Replace", "modern, dark wood nightstand", "simple, white two-drawer nightstand", ("near", "white cabinet")]
   ```
- **Translate the nightstand to ensure symmetry with another nightstand near the bed:**
   ```json
   ["Translate", "simple, white two-drawer nightstand", "x", 1.22]
   ```
Combining these together, the final output is:
**Output:**
***["Replace", "modern, dark wood nightstand", "simple, white two-drawer nightstand", ("near", "white cabinet")];
["Translate", "simple, white two-drawer nightstand", "x", 1.22]***

Figure 6: Dialog between user and LLM to generate an edit plan to rearrange the bedroom. The room image in 'User Input' is not given to LLM, just for illustration purpose.

**System Prompt**

Given scene configurations and templated commands, you should write new commands using natural language and spatial reference.
Templated commands will be in the 'action: target_object' format. If the location is provided in the templated commands, it can be considered as a hint for the target object's location compared to the existing object in the scene.
All sizes and centroids in scene configurations are in meters. The angles are defined in degrees. The dimension sequence is [x,y,z]. Vertical angles are the angles along the y-axis.
Sizes are the half lengths of the bounding box along the x, y, and z axes when the vertical angle is zero.
We define +x/-x as the right/left direction, +y/-y as the up/down direction, and +z/-z as the front/back direction.
When you design new commands, please refer to the spatial relations between objects in the scene.
When you design new commands, please consider correctness, conciseness, and naturalness.
You should attempt to make your command need reasoning.
If there are duplicate target objects in the scene, you should refer to object locations by relative spatial relations with one unique object in the scene.
If there are multiple templated commands, you should consider them as the same command with different representations.
If templated commands indicate to add an object where there is already a similar object, you should indicate this is about adding a new object in your command.
Enlarge and shrink in the command should be uniform.
You can add object descriptions according to the scene configurations, commands, and the image (if provided).
For example:
Example1:
[Templated commands]:['move object towards the ***left*** direction for 1 meters: a white bed with a red and white plaid comforter and a red and white plaid pillow.']
If there is a table (only one table inside the scene) on the left side of the bed and length of bed is 2 meters, you can write: 'move the white bed with red and white plaid towards the table around 1 meters.' or 'move the bed towards the left direction by half of bed length.'
Example2:
[Templated commands]:['move object towards the ***left*** direction for 0.5 meters: a wooden nightstand.']
If there is a bed parallel to the nightstand and moving to the left will make the nightstand closer to the bed headboard, you can write: 'move the nightstand closer to the bed headboard by 0.5 meters'.
Example3:
[Templated commands]:['replace source with target : [Source] a white bed; [Target] a brown bed.']
You can write: 'replace the white bed with a brown bed.'
Example4:
[Templated commands]:['add object: a white bed; location: ***right*** a wardrobe.']
If there is a wardrobe in the scene, you can write: 'add a white bed on the right side of the wardrobe.'
Now you can start to design new commands based on the scene configurations and templated commands. You can supplement object descriptions on the command.
Think about it step by step and summarize your commands in the end. The final output format should be '###[natural command 1, natural command 2, ...]###', which is a list of strings and can be processed by ast.literal_eval() or json.loads().

---

**User Input**

[Scene configurations]:
Object 0: {"class": "dining table", "size": [0.55, 0.38, 0.23], "vertical angle": 0, "centroid": [-0.8, 0.0, -3.46],
"description": "the dining table is a modern, minimalist design with a black marble top and a silver metal frame."}
Object 1: {"class": "loveseat sofa", "size": [1.24, 0.43, 0.47], "vertical angle": 90, "centroid": [-3.78, 0.0, 1.38],
"description": "the loveseat sofa is brown with a modern design and has a variety of patterned throw pillows."}
Object 2: {"class": "coffee table", "size": [0.69, 0.23, 0.47], "vertical angle": 90, "centroid": [-2.48, 0.0, 1.4],
"description": "the coffee table is a modern, minimalist design with a geometric shape, featuring a combination of dark wood and lighter wood panels."}
Object 3: {"class": "lounge chair", "size": [0.37, 0.45, 0.37], "vertical angle": 153, "centroid": [-2.94, 0.0, 3.11],
"description": "the chair is a modern, minimalist design with a dark wood frame and a cushion featuring a geometric pattern."}
Object 4: {"class": "corner side table", "size": [0.22, 0.23, 0.22], "vertical angle": 90, "centroid": [-3.99, 0.0, -0.25],
"description": "a round, black marble table with a white base."}
Object 5: {"class": "dining chair", "size": [0.31, 0.45, 0.3], "vertical angle": -180, "centroid": [-0.47, 0.0, -2.79],
"description": "the chair is black with a modern design, featuring a high back and armrests."}
Object 6: {"class": "dining chair", "size": [0.31, 0.45, 0.3], "vertical angle": -180, "centroid": [-1.08, 0.0, -2.79],
"description": "the chair is black with a modern design, featuring a high back and armrests."}
Object 7: {"class": "corner side table", "size": [0.22, 0.23, 0.22], "vertical angle": 90, "centroid": [-3.99, 0.0, 3.13],
"description": "a round, black marble table with a white base."}
Object 8: {"class": "dining chair", "size": [0.31, 0.45, 0.3], "vertical angle": 0, "centroid": [-1.08, 0.0, -4.17],
"description": "the chair is black with a modern design, featuring a high back and armrests."}
Object 9: {"class": "dining chair", "size": [0.31, 0.45, 0.3], "vertical angle": 0, "centroid": [-0.48, 0.0, -4.17],
"description": "the chair is black with a modern design, featuring a high back and armrests."}
Object 10: {"class": "console table", "size": [0.7, 0.42, 0.15], "vertical angle": 0, "centroid": [-3.35, 0.0, -4.56],
"description": "the console table is a modern, black, three-tiered design with a flat top and a rectangular base."}
Object 11: {"class": "cabinet", "size": [0.62, 1.08, 0.29], "vertical angle": -90, "centroid": [-3.94, 0.0, -2.04],
"description": "the children's cabinet is a modern, minimalist design with a light wood frame and a blue and white color scheme, featuring a playful bunny motif on the doors."}
Object 12: {"class": "pendant lamp", "size": [0.18, 0.54, 0.18], "vertical angle": 90, "centroid": [-2.71, 1.53, 4.35],
"description": "the pendant lamp is a modern, metallic chandelier with a white finish, featuring a series of vertical, clear glass tubes that create a geometric pattern."}
Object 13: {"class": "pendant lamp", "size": [0.19, 0.52, 0.19], "vertical angle": 90, "centroid": [-2.26, 1.55, -0.3],
"description": "the pendant lamp is black with a woven design and a white interior."}

[Templated commands]:["rotate object 135 degrees : the pendant lamp is black with a woven design and a white interior",
"obviously rotate object 135 degrees :the pendant lamp is black with a woven design and a white interior."]
Hint: The target object is the Object_13.

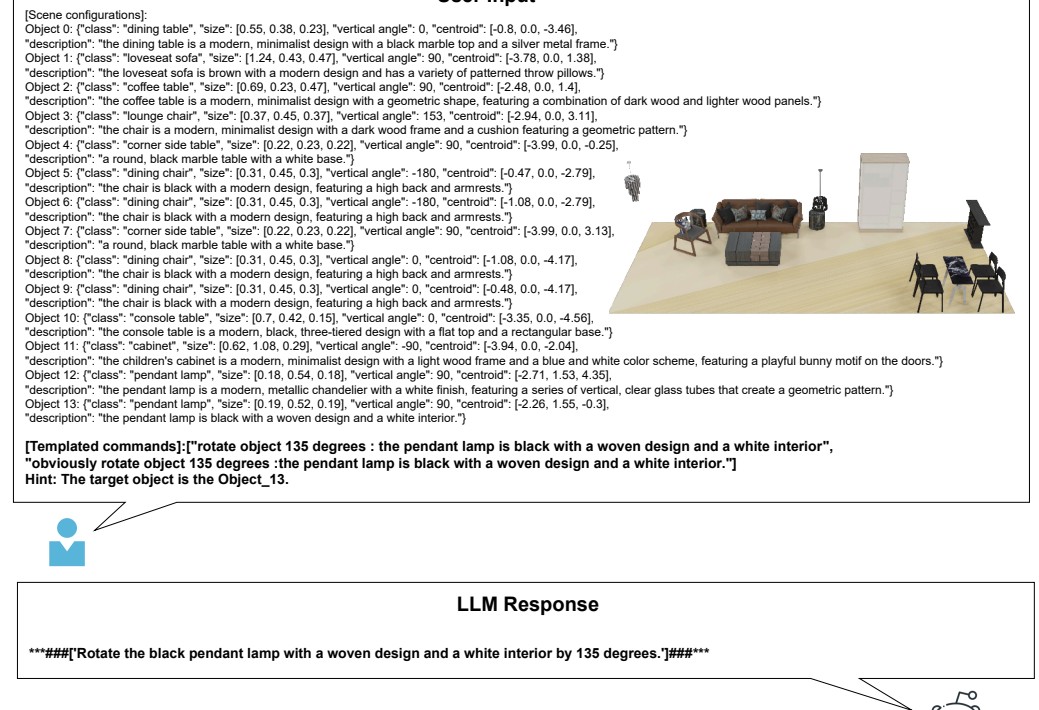

---

**LLM Response**

***###['Rotate the black pendant lamp with a woven design and a white interior by 135 degrees.']###***

Figure 7: Dialog between user and LLM to generate natural language description from the template command. The room image in 'User Input' is not given to LLM, just for illustration purposes.

