# OpenReview forum: "EditRoom: LLM-parameterized Graph Diffusion for Composable 3D Room Layout Editing"
_ICLR.cc/2025/Conference — ICLR 2025 Poster_

### Official Review · Reviewer_d67d · 2024-10-30

**Soundness:** 2
**Presentation:** 3
**Contribution:** 2
**Rating:** 6
**Confidence:** 4

**Summary:**

This paper proposes EditRoom for 3D layout editing of an existing room. It first uses GPT-4o to parse the natural language instruction into a sequence of breakdown commands, and then trains a diffusion model to generate the edited layout based on the commands. The framework is reasonable and utilize the state-of-the-art LLM and generative diffusion model. The proposed approach generates good results and outputs the other variants of itself, validating the effectiveness of using GPT-4o to infer the breakdown commands and the design of the graph diffusion model as 3D scene editor.

**Strengths:**

The strengths include:

1. It proposes the 3D layout editing pipeline, which uses GPT-4o as the command parameterizer and a graph diffusion model as the scene editor.

2. The graph diffusion is tailored to deal with the layout editing problem and outperforms the adapted DiffuScene approach.

3. It constructs a large dataset composed of the paired scene layouts and the corresponding editing instructions, which facilitates the following language-based 3D layout editing research.

**Weaknesses:**

**Firstly, it lacks the discussion of the motivation for the target problem: why and when a user would prefer a language-based 3D layout editing rather than interactively modify the one or two object placements by himself?** This relates with the design of evaluation experiments. Intuitively, the problem setting should be a complicated editing and a very rough instruction. For example, the user might not explicitly specify all the atom editing, but rather describe the key intent only. Or it requires a series of operations to make enough space to enable the target editing. The single-operation and multi-operation cases shown in the paper seem straightforward. Actually, the second row  in Fig.4 gave a good example, since a user might express “rotate the bed” but desire to have the bed s well as the accompanied nightstand rotated together. But it seems that the proposed approach only edit the objects that are explicitly specified, and produces an unreliable layout result.

**Secondly, it lacks the comparison with heuristic methods.** The proposed approach trains a graph diffusion model as the scene editor. But the paper only compare with DiffuScene (another diffusion model for 3D room layouts) and its own variants, only to validate the key designs such as taking breakdown commands as conditions and the condition types of the network. In fact, training neural networks for 3D layout problem often aims to deal with high-level conditions and efficiently generate plausible layouts. Given that the GPT-4o has already inferred the breakdown commands and the generative process of diffusion model is slow, do we really need such a network to produce the edited layout? What if directly parse the breakdown commands to modify the specified object? What if using a simulated annealing algorithm as a post-processing step to deal with the possible collisions? At least it alleviates the demand for constructing a training dataset.

**Thirdly, it lacks a detailed discussion of the command parameterizer.** Currently, there are different opinions on whether the LLM has satisfactory spatial reasoning ability. Fig.5 is a good example as the goal is to make the two nightstands symmetric with each other. But it seems the LLM response presented in the figure is incomplete. It is important to give a in-depth and detailed analysis on the LLM performance, e.g. when the LLM would fail and how to promote its ability?

There are some minor issues. For example, are the point cloud of each object (line 227) only point locations or rgb color as well? How to “concatenate each element of source scene graphs into noisy target scene graphs” (line 241)? It means to concatenate the attributes of corresponding elements, or merge the node lists?

**Questions:**

Overall, this paper presents a novel framework for language-based 3D room layout editing. The main weaknesses lie in the motivation and evaluation of the proposed approach. The current evaluation only validate the key designs under the novel framework (first LLM and then a diffusion model), but lacks the explanation on why this framework is essential (especially the diffusion model as a scene editor).

---

> ### Author Response · Authors · 2024-11-22
> **Response to Reviewer d67d (1/2)**
>
> We appreciate the reviewer’s acknowledgment of the method design and dataset. We are willing to address the concerns as follows:
> 1. **(W1) Motivation of problem.** We would like to clarify that the primary goal of our language-based 3D editing model is not to perfectly execute every editing command in a single attempt. Instead, it aims to lower the barrier for non-expert users who may lack the technical proficiency or time to manipulate 3D scenes interactively, particularly for operations involving replacement or adding objects.
>
>     When users attempt to add a new object to a scene, they often face the time-consuming process of searching through massive 3D asset databases to find a suitable object. After selecting an object, further effort is required to import it into the scene, adjust its scale, and position it correctly—all of which demand both significant time and proficiency with 3D editing software. These challenges can be daunting for users unfamiliar with such tools.
>
>     To address these issues, our approach focuses on making 3D layout editing more efficient and accessible. By conditioning on the user’s natural language prompts, our model generates reasonable initial editing results that serve as a foundation for further refinement. Importantly, the scenes remain fully editable, allowing users to either provide additional prompts for iterative improvements or manually make minor adjustments to achieve their desired outcome.
>
>     In summary, our work is designed to enhance efficiency, streamline the editing process, and empower users with an intuitive way to interact with 3D layouts, without requiring extensive technical expertise. We appreciate the reviewer’s feedback and have incorporated this perspective more explicitly in the revision.
>
> 2. **(W2) Comparison with heuristic methods.** We appreciate the reviewer’s suggestion to explore heuristic methods and acknowledge their potential for addressing specific 3D scene editing tasks. However, we believe that our graph diffusion model offers significant advantages over heuristic approaches, particularly in terms of flexibility, scalability, and efficiency.
>
>     **First**, heuristic methods require manually designing numerous rules to handle different operations such as adding, removing, replacing, and adjusting objects. These rules need to account for a wide variety of scenarios and spatial relationships, making them difficult to generalize across diverse editing tasks. In contrast, our diffusion model learns these patterns directly from data, eliminating the need for labor-intensive rule creation and enabling adaptability to new or complex scenarios.
>
>     **Second**, when operations involve adding a new object to a scene, heuristic methods such as simulated annealing must optimize multiple parameters, including size, position, and rotation, to ensure the object fits harmoniously into the layout. This process is computationally expensive and often fails to account for the global context of the scene. For example, while the algorithm might find a collision-free placement, the result may not align aesthetically or functionally with the surrounding objects. Our diffusion model inherently considers these factors, producing edits that are both context-aware and harmonious.
>
>     **Third**, heuristic methods often produce deterministic outputs, which can limit diversity and creativity in scene editing. Our diffusion model can generate a range of plausible layouts, providing users with more options to refine their designs.

---

> > ### Author Response · Authors · 2024-11-22
> > **Response to Reviewer d67d (2/2)**
> >
> > 3. **(W3) Discussion of command parameterizer.** During our experiments, we observed that LLMs may fail to generate precise geometric values, such as translation offsets or rotation angles. Since how to improve LLM’s spatial reasoning ability is still an open question and is not the primary focus of our work, we utilize diffusion models to mitigate the influence of potential erroneous commands.
> >
> >     There is an inherent trade-off between strictly adhering to the commands and avoiding inappropriate operations. Diffusion models, trained on clean datasets, inherently learn the overall scene distributions and prioritize generating edits that align with these distributions. This ensures that the outputs remain contextually appropriate and coherent, even when the input commands contain inaccuracies.
> >
> >     In such cases, the LLM's role is to provide a high-level intent or editing direction, while the diffusion model refines the execution to maintain spatial and relational plausibility. This collaborative approach allows us to combine the strengths of both systems: the semantic understanding of the LLM and the generative precision of the diffusion model.
> >
> > 4. **(W4) Clarification of point cloud.** Following the setting of the pretrained OpenShape encoder, all point clouds include both locations and RGB colors.
> >
> > 5. **(W5) Clarification of source scene injection.** In this work, we merge the source scene nodes to the target scene nodes and expand source scene edges to the target scene edges as context. The edges across the source scene nodes and target nodes have been set to dummy values.

---

> > > ### Comment · Reviewer_d67d · 2024-11-25
> > >
> > > Although I'm still worried about the motivation of the single-operation scene edits, I appreciate the authors' response and the multi-operation results presented in the paper. I suggest the authors to consider performing a light-weight optimization to refine the results. I have increased my rating.

---

> > > > ### Author Response · Authors · 2024-11-25
> > > > **Response to Reviewer d67d**
> > > >
> > > > We sincerely thank the reviewer for their thoughtful feedback, increased rating, and acknowledgment of our multi-operation results. Incorporating heuristic methods to refine the results is an excellent suggestion, and we are willing to explore this approach in future revisions to enhance our framework further.

---

### Official Review · Reviewer_SwhT · 2024-11-02

**Soundness:** 3
**Presentation:** 4
**Contribution:** 3
**Rating:** 8
**Confidence:** 4

**Summary:**

This paper introduces EditRoom, a two-module scene editing framework based on natural language prompts. First, the command parameterizer converts natural language prompts into sequences of breakdown commands for six editing types using an LLM. Then, the scene editor constructs the target scene from the source one using the proposed conditional graph generation models based on diffusion. Quantitative and qualitative results prove the effectiveness and generalization capability of the proposed approach.

**Strengths:**

a) The problem studied in the paper is important for real-world applications. b) This paper introduces a synthetic scene editing dataset called EditRoom-DB, contributing to the field. c) Support for multi-operation commands enhances the practicality of the proposed approach.

**Weaknesses:**

a) Comparison with baselines: The authors mention the difficulty of adapting scene synthesis methods for layout editing, and thus compare with only two baselines. However, several LLM-based methods using in-context learning (e.g. LayoutGPT) could potentially be adapted to support editing by defining the task for the LLM and conditioning on in-context learning samples from EditRoom-DB. Including such a comparison with in-context learning-based methods would enhance the evaluation. b) Missing complexity analysis: In this work, the authors train two graph diffusion models, however the complexity analysis is missing. How much time do the training and inference take?

**Questions:**

All of my questions are listed in the weaknesses section, and I may adjust the rating if they are well addressed.

---

> ### Author Response · Authors · 2024-11-22
> **Response to Reviewer SwhT**
>
> We appreciate the reviewer’s acknowledgment of the motivation, dataset, and methods. We would like to address the concerns as follows:
> 1. **(W1) Comparison with LayoutGPT.** We appreciate the suggestion of including LLM-based methods like LayoutGPT as baselines. We would like to mention that LayoutGPT does not establish a text-based object retrieval pipeline, which limits its ability to add or replace objects in a scene and preserve source object identities. Following the reviewer’s suggestion, we constructed scenes as prompts and randomly sampled editing examples from the training dataset to serve as in-context examples. Due to the associated time and computational costs, we evaluated LayoutGPT on the bedroom test set containing 4,800 samples. The results are presented below:
>
>     |               | **Model**       | **IOU (↑)** | **S-IOU (↑)** | **LPIPS (↓)** | **CLIP (↑)** | **IOU (↑)** | **S-IOU (↑)** | **LPIPS (↓)** | **CLIP (↑)** |
>     |---------------|-----------------|-------------|---------------|---------------|--------------|-------------|---------------|---------------|--------------|
>     | **Operation** |                 | Translate   |               |               |              | Rotate      |               |               |              |
>     |               | **LayoutGPT**  | 0.6734      | 0.6432        | 0.1549        | 0.9402       | **0.7978**      | 0.7554        | 0.1115        | 0.9433
>     |
>     |               | **EditRoom**        | **0.6877**  | **0.6831**    | **0.1256**    | **0.9636**   | 0.7654  | **0.7585**    | **0.0856**    | **0.9685**   |
>     | **Operation** |                 | Scale       |               |               |              | Replace     |               |               |              |
>     |               | **LayoutGPT**   | 0.7734      | 0.7205        | 0.1273        | 0.9304       | 0.7122      | 0.6697        | 0.1526        | 0.9207       |
>     |               | **EditRoom**        | **0.7781**  | **0.7724**    | **0.0790**    | **0.9728**   | **0.7286**  | **0.7018**    | **0.1327**    | **0.9418**   |
>     | **Operation** |                 | Add         |               |               |              | Remove      |               |               |              |
>     |               | **LayoutGPT**   | 0.6424      | 0.6125        | 0.1829        | 0.9183       | **0.7976**      | 0.7547        | 0.0911        | 0.9495       |
>     |               | **EditRoom**        | **0.6679**  | **0.6566**    | **0.1553**    | **0.9407**   | 0.7716  | **0.7644**    | **0.0789**    | **0.9698**   |
>
>     From these experiments, we find that LayoutGPT demonstrates a reasonable ability to maintain the original layout of the source scene and handle some basic low-level operations, like rotation and scaling. However, LayoutGPT may execute wrong editing operations. We further calculate the **target object IOU** by only considering the target object overlapping in the generated scenes, where EditRoom is **0.52** and LayoutGPT is **0.41**, which indicates EditRoom has more accurate editing capability. Besides, since LayoutGPT retrieves objects only based on category and bounding box size, its outputs lack finer control and accuracy. This limitation is evident in the lower S-IOU scores and image similarity metrics (LPIPS and CLIP) compared to EditRoom. We believe combining LayoutGPT's layout maintaining ability and EditRoom's editing ability can be an interesting direction for further research.
>
> 2. **(W2) Training and inference time.** Each diffusion model was trained for 300 epochs using 8 NVIDIA A5000 GPUs (24 GB each), with the training process taking approximately 2 hours per model. For inference, we set the diffusion denoising steps to 100, resulting in a runtime of approximately 5 seconds per diffusion process.

---

> > ### Comment · Reviewer_SwhT · 2024-11-25
> >
> > Thank you for responding to my questions, the rebuttal mostly addressed my concerns.
> >
> > Although I appreciate the comparison with an in-context learning based method, this comparison could potentially be enhanced by selecting in-context samples based on the edit type from EditRoom-DB. I suggest the authors to consider this to further strengthen the comparison.

---

> ### Author Response · Authors · 2024-11-25
> **Response to Reviewer SwhT**
>
> We sincerely thank the reviewer for their prompt reply and valuable feedback. We apologize for not providing sufficient details regarding the in-context example selection strategy, which may have caused some misunderstanding. **To clarify, yes, the results presented above were obtained using editing samples based on the same edit type from EditRoom-DB as the in-context examples.**

---

> > ### Author Response · Authors · 2024-12-02
> > **Response to Reviewer SwhT**
> >
> > Dear Reviewer SwHT,
> >
> > Thank you again for your constructive feedback! Since our results of LayoutGPT were indeed obtained using in-context examples from EditRoom-DB with the same edit type, we believe all your concerns and suggestions are addressed and incorporated now.
> >
> > Therefore, we kindly wonder if you can increase your rating to reflect it and support our work. We’re also happy to hear any additional suggestions to further improve the paper. Thank you for your time and thoughtful review!
> >
> > Best,
> >
> > Authors of Submission 12972

---

### Official Review · Reviewer_RttZ · 2024-11-03

**Soundness:** 3
**Presentation:** 3
**Contribution:** 3
**Rating:** 6
**Confidence:** 4

**Summary:**

Given a source 3D scene and a natural language command describing the scene edits, the paper presents a method to generate the  corresponding 3D edited scene.

The existing gaps that motivate this paper are three folds: (a) Requiring manual intervention to perform 3D scene edits, (b) limited number of edits and a lack of support for multi-operational textual commands, and (c) lack of a language-guided scene editing dataset, making it difficult to train and evaluate scene-editing models.

To this end, the contributions of the paper include proposing a new approach that combines an LLM (to encode edit commands) with a graph diffusion model (to incorporate scene edits from the input language command) to effectively serve the high-level goal of synthesizing a 3D scene based on natural-language edits/commands. The framework is termed “EditRoom”. In addition to the proposed approach, the paper also introduces a new synthetic scene editing dataset named “EditRoom-DB” with 83K pairs of language+edited 3D scene, spanning 16K initial 3D scenes.

To summarize, EditRoom takes as input a source scene and a natural language command describing scene edits, and outputs a corresponding edited scene. The dataset used is “EditRoom-DB”, which is a new dataset of scene editing pairs created from 3D-FRONT  dataset. GPT-4o is the LLM used, and there is a graph-transformer based diffusion model. The learning setup is strongly supervised. The loss functions involved are KL divergence loss and diffusion loss for graph diffusion model and layout diffusion model. In terms of evaluations, the paper looks at the following four quantitative metrics: (1) IoU score: 3D IoU between objects in pre-edit and post-edit scene, and selecting pairs with highest 3D IoU value (Q: how many pairs do you use?) (2) S-IoU score: semantic similarity between captions of matching objects as measured by Sentence BERT (Q: how do you find caption for generated objects?) (3) LPIPS: determines pixel similarity between 24 views of pre-edit and post-edit scenes, and (4) CLIP score: Using CLIP-ViT-B32 to find semantic similarity between 24 views. As for comparisons, the paper compares against two works: DiffuScene-N (modified version of existing 3D scene synthesis model DiffuScene, from CVPR 2024), and  ScenEditor-N. (Spelling mistake: Line 323, the name of the baseline should be DiffuScene-N, instead of DiffuScene-E)

**Strengths:**

1) Well-written paper
2) The idea of breaking down command into multiple steps is interesting
3) The proposed model can be used for multi-operation editing without explicitly training
on such commands. This is a useful feature for interior designers
4) The newly introduces dataset “EditRoom-DB” is quite useful to the 3D scene research community

**Weaknesses:**

1) The motivation to use diffusion model is unclear.
2) Out of the two compared methods, only one, DiffuScene-N, is based on an existing work (viz., DiffuScene, from CVPR 2024, Tang et. al.). The other one is a modified version of the proposed approach. This is good, But, why not use a modified version of InstructScene (Haque et. al., ICLR 2024) and also of EchoScene (Zhai et.al. , ECCV 2024) as well? This would be tell a good story on how effective the proposed approach is. This is a weakness, something I believe can be addressed.

**Questions:**

1) Is the scene edit one-shot (i.e., take all edit commands at once and generate a scene) or progressive (i.e., edit the 3D scene one command at a time)?
2) Motivation behind using a diffusion model is unclear. If the edits allowed are simply adding,
removing, replacing, scaling, translating and rotating an object (even the relations
between objects are pre-defined and limited), why do you need to train a diffusion
model? You could perform these tasks directly on the source scene graph by
adding/removing/replacing a node and then retrieving an object from the given object
database that fits the text description (see Language-Driven Editing of 3D Scenes from Scene Databases, SIGGRAPH 2018; missing this reference). Using diffusion model makes sense when one
wants to generate an entirely new scene/new object from scratch. In the scene editing
task, we are given with an initial scene and our task is to modify it.
3) How does the diffusion process look like for, say, rotation operation? It would be
helpful to have visualizations of intermediate diffusion steps. Applying rotation to
existing object can be done analytically. Why would we need a diffusion model? I would consider this a weakness of the paper as it precludes some important visualizations pertaining to the proposed approach and something that is relevant to the end results.
4) Regarding the proposed dataset “EditRoom-DB”, what is the reason behind having only
500 examples as test data? This number is too small compared to the number of
training samples i.e. 83k. The standard practice is to have a certain percentage (say,
10%) of total samples as test data. Looking at Table-1, dataset in its current form is
biased towards “bedroom” category because it has ~50k training and only 500 test
samples, as compared to ~18k train and 500 test samples for “dining room” category.
The ratio of train-test data is high for bedroom scenes compared to the other two categories, and its
effect is seen in the results Table-2 (high performance for bedroom than dining and
living room). It would be a good experiment to run when you keep the train:test ratio
equal for all categories and see if the model gives comparable results.

Overall, the paper proposes a novel 3D indoor scene editing model that uses LLM to encode
scene editing instruction and a graph-diffusion model to perform scene edits. Moreover, it introduces a
new dataset for training and evaluating 3D indoor scene editing task.

---

> ### Author Response · Authors · 2024-11-22
> **Response to reviewer RttZ (1/2)**
>
> We appreciate the reviewer’s acknowledgment of the motivation, paper presentation, and methods. We are willing to address the concerns in the following:
> 1. **(W1,Q2) Motivations for using diffusion.** Thanks for the reminder of the missing related work [1], we have updated it in the revision. There are three main motivations for using diffusion models instead of rule-based methods (like [1]) in our work.
>
>     **First**, the diffusion model can enhance diversity in outputs. Unlike rule-based methods, which tend to produce deterministic results, the diffusion process introduces controlled randomness during generation. This diversity is particularly beneficial for operations like adding or replacing objects, where multiple plausible configurations could fit the given command.
>
>     **Second**, the diffusion models are more scalable, generalizable, and efficient. Diffusion models inherently consider the entire scene layout distribution, enabling them to generate edits that align with the broader spatial and relational context of the scene. For instance, when adding a wardrobe next to a table, the diffusion model does not only account for the table’s pose but also evaluates the overall layout to ensure the new configuration is coherent and harmonious. In contrast, rule-based methods require extensive hand-crafted rules to cover edge cases, making them less scalable and less adaptable to complex scenes. Diffusion models, by learning directly from the whole scene distribution, can generalize across diverse layouts and generate appropriate configurations without additional manual intervention. Furthermore, finding collision-free configurations in rule-based methods is often time-consuming, involving iterative checks and adjustments. Diffusion models, as end-to-end frameworks, can directly generate reasonable configurations in one pass, significantly improving efficiency.
>
>     **Third**, diffusion models are highly effective at incorporating textual guidance into the generation process. Instead of using complex language parsers in rule-based methods, the diffusion process can align its outputs with the semantic intent of commands through the cross-attention mechanism.
>
>     Besides, diffusion models are also suitable for editing tasks. Like the changes from Stable Diffusion [2] to InstructPix2Pix [3] in the image editing domain, where the source images are concatenated as context, we adopt a similar idea by incorporating source scene graph as context for 3D scene layout editing. After training, the diffusion models can learn what properties should be preserved, and others need to be changed according to the commands.
>
> 2. **(W2) Clarification of baselines.** We would like to clarify that SceneEditor-N can be considered a modified version of InstructScene. The original InstructScene framework does not accept source scenes as input, and its language conditions are primarily incorporated for semantic scene graph generation, which restricts its ability to perform fine-grained, low-level layout control required for editing tasks. To address these challenges, we developed SceneEditor, which builds on InstructScene by introducing significant structural updates to enable effective 3D scene editing.
> Regarding EchoScene, its primary focus is on text-free layout generation, where semantic scene graphs are provided as input for generating layouts. Modifying EchoScene to generate new scene graphs and layouts with text conditions would require significant revisions to its core framework, making it less suitable as a baseline for our work.
>
> 3. **(Q1) Editing execution sequence.** For multiple breakdown commands, we execute each command progressively.
>
> [1] Ma, Rui, et al. "Language-driven synthesis of 3D scenes from scene databases." ACM Transactions on Graphics (TOG) 37.6 (2018): 1-16.
>
> [2] Rombach, Robin, et al. "High-resolution image synthesis with latent diffusion models." Proceedings of the IEEE/CVF conference on computer vision and pattern recognition. 2022.
>
> [3] Brooks, Tim, Aleksander Holynski, and Alexei A. Efros. "Instructpix2pix: Learning to follow image editing instructions." Proceedings of the IEEE/CVF Conference on Computer Vision and Pattern Recognition. 2023.

---

> ### Author Response · Authors · 2024-11-22
> **Response to reviewer RttZ (2/2)**
>
> 4. **(Q3) Visualization of intermediate diffusion steps.** Thanks for the suggestion regarding the visualization of intermediate diffusion steps. We have added these as Figure 5 in the appendix of the revision. At the start of the layout diffusion process, the poses of all objects begin as random noise. As the diffusion process progresses, each object's pose is iteratively refined and placed into an appropriate configuration that aligns with the source scene and the given commands. This iterative refinement highlights the strength of the diffusion model in handling scene edits cohesively. By adopting a unified approach, the diffusion model is capable of jointly generating the entire scene layout for all editing types, ensuring consistency and coherence across diverse operations.
>
> 5. **(Q4) Balance train:test ratio.** Thanks for the suggestion of balancing train:test ratio for different room types. We have re-sampled the test sets (bedroom 4800, livingroom 1500, and dining room 1500) to make the train:test ratios close to 10% for each room type, and the results are shown below:
>
>     |  Model  | **Bedroom** |          |       |         | **Dining Room**|          |       |   | **Living Room**|          |       |       |
>     |------------------|---------|-----------|-----------|----------|---------|-----------|-----------|----------|---------|-----------|-----------|----------|
>     |           | IOU (↑) | S-IOU (↑) | LPIPS (↓) | CLIP (↑) | IOU (↑) | S-IOU (↑) | LPIPS (↓) | CLIP (↑) | IOU (↑) | S-IOU (↑) | LPIPS (↓) | CLIP (↑) |
>     | **DiffuScene-N** | 0.6220  | 0.6125    | 0.1431    | 0.9415   | 0.4473  | 0.5196    | 0.1921    | 0.9254   | 0.4332  | 0.4161    | 0.1810    | 0.9236   |
>     | **SceneEditor-N**| 0.7055  | 0.6942    | 0.1261    | 0.9510   | 0.5262  | 0.5091    | 0.1557    | 0.9366   | 0.4618  | 0.4498    | 0.1738    | 0.9329   |
>     | **EditRoom**     | **0.7342**| **0.7236**| **0.1090**| **0.9597**| **0.5360**| **0.5196**| **0.1460**| **0.9460**| **0.4710**| **0.4629**| **0.1616**| **0.9446**|
>
>     From the results, we find that the conclusions are still unchanged. We have updated the related results in the revision.
>
> 6. **(Q1 in summary) Clarification of 3D IOU. For calculating 3D IOU,** we iterate all objects in the ground truth target scene to find corresponding objects with the largest IOU in the generated scene. If one ground truth object does not find the overlap object in the generated scene, its 3D IOU will be zero. Then, we average all 3D IOU scores of target scene objects as the score for the editing results.
>
> 7. **(Q2 in summary) Clarification of object captions.** In our work, all objects are retrieved from a pre-collected object dataset (3D-FUTURE). Therefore, we used LLAVA-1.6 to pre-generate all object captions, mentioned in section 4 of the paper.

---

> > ### Comment · Reviewer_RttZ · 2024-11-23
> >
> > Authors, thank you for responding to my questions.
> >
> > The rebuttal addresses most of the questions raised in my review.
> >
> > In your rebuttal point 2 (W2, Clarification of Baselines), I did not see any talking point about EchoScene from ECCV 2024. That is a good paper to look at for SOTA comparison -- The code is out (https://github.com/ymxlzgy/echoscene) and it should not be difficult to make a comparison. Let us try to have a comparison to EchoScene to further strengthen the paper.
> >
> > For now, this is the one improvement that I see helping the paper. Please keep us updated.

---

> > > ### Author Response · Authors · 2024-11-23
> > > **Response to reviewer RttZ**
> > >
> > > We sincerely thank the reviewer for their quick reply and thoughtful discussion. We recognize that EchoScene represents the current state-of-the-art for 3D scene generation. However, we would like to respectfully clarify that EchoScene is not directly suitable for language-guided 3D editing tasks without significant modifications. Below, we outline two main challenges:
> > >
> > > 1. **Focus on Graph-Conditioned Layout Generation**:
> > >
> > >     EchoScene primarily focuses on layout generation using pre-established scene graphs, rather than generating layouts directly from text. Specifically, the input to EchoScene consists of scene graphs, and the output is a corresponding 3D layout. For reference, we quote the description from Appendix E.1 (end of page 24) in the original EchoScene paper:
> > >
> > >     > **"Our task focuses on graph-conditioned scene synthesis. Therefore, we conduct the experiment solely in the second stage of InstructScene."**
> > >
> > >     In InstructScene, there are two stages: (1) text-to-graph generation and (2) graph-to-layout generation, similar to the two diffusion models used in EditRoom. EchoScene explicitly mentions that it can only compare with the second stage of InstructScene. However, in language-guided 3D editing tasks, the second stage cannot function independently, as operations like adding, removing, or replacing objects require modifications to the scene graph itself.
> > >
> > > 2. **Lack of Text Input Support**:
> > >
> > >     EchoScene does not incorporate textual input for its layout generation. This is evident from the methodology and illustrations in the original paper (Figures 1, 2, and 3), where no text encoder or instruction-processing mechanism is included. In their qualitative results, all scene graphs are manipulated manually. Consequently, EchoScene is fundamentally limited in its ability to handle language-guided tasks, which rely on understanding and executing textual descriptions.
> > >
> > > We appreciate the reviewer’s helpful suggestions and have updated the revision to explicitly highlight the differences between EchoScene and EditRoom, making their distinct capabilities and focus areas clearer.

---

> > > > ### Comment · Reviewer_RttZ · 2024-11-28
> > > >
> > > > Thank you for the clarification on EchoScene. Indeed, EchoScene does graph-to-layout generation, without supporting text input, which would be a divergent comparison.
> > > >
> > > > Highlighting these differences is a good thing (which you have done).
> > > >
> > > > With that, my concerns have been addressed. I would like to keep my score (6, above Acceptance threshold) looking at the overall contributions of the paper.

---

> > > > > ### Author Response · Authors · 2024-11-28
> > > > > **Response to Reviewer RttZ**
> > > > >
> > > > > Thank the reviewer for their thoughtful feedback and for acknowledging the addressed concerns. We appreciate their constructive comments, which have significantly helped improve our work. We are glad that the revisions align with the expectations. Thanks for your time and support throughout the review process!

---

### Comment · Area_Chair_5J2G · 2024-11-25
**Please read the rebuttal and reply**

Dear Reviewers,

Thanks again for serving for ICLR, the discussion period between authors and reviewers is approaching (November 27 at 11:59pm AoE), please read the rebuttal and ask questions if you have any. Your timely response is important and highly appreciated.

Thanks,

AC

---

### Meta-Review · Area_Chair_5J2G · 2024-12-23

**Metareview:**

This paper proposes a method that provides a unified framework for language-guided 3D scene layout editing.The main idea is to leverage a LLM and a graph diffusion-based model to edit 3D scenes represented as semantic graphs and layouts. Other contributions include a new 3D dataset for the editing task.

The proposed method resolves an important and practical task with an interesting solution. During rebuttal, the reviewers raised issues including motivation, method, datasets, missing comparisons and efficiency performance. After rebuttal, all reviewers agree to accept this paper.

**Additional Comments On Reviewer Discussion:**

During rebuttal, the reviewers raised concerns including:
- Motivation of the target task as well as using diffusion models (Reviewer RttZ, d67d)
- Technical questions about the proposed method and dataset (Reviewer RttZ, d67d)
- Missing comparison with baselines (Reviewer SwhT, d67d)
- Missing efficiency performance (Reviewer SwhT)

All reviewers agree to accept this paper after rebuttal.

---

### Decision · Program_Chairs · 2025-01-22

Accept (Poster)